# Cancer Is Associated with Alterations in the Three-Dimensional Organization of the Genome

**DOI:** 10.3390/cancers11121886

**Published:** 2019-11-27

**Authors:** Lifei Li, Nicolai K. H. Barth, Christian Pilarsky, Leila Taher

**Affiliations:** 1Division of Bioinformatics, Department of Biology, Friedrich-Alexander-Universität Erlangen-Nürnberg, 91058 Erlangen, Germany; lifei.li@fau.de (L.L.); nicolai.barth@fau.de (N.K.H.B.); 2Department of Surgery, Friedrich-Alexander-Universität Erlangen-Nürnberg and Universitätsklinikum Erlangen, 91054 Erlangen, Germany; christian.pilarsky@uk-erlangen.de; 3Institute for Biomedical Informatics, Graz University of Technology, 8010 Graz, Austria

**Keywords:** cancer, the cancer genome atlas (TCGA), topologically associating domains (TADs), copy number variants (CNVs), LASSO cox regression analysis, biomarkers

## Abstract

The human genome is organized into topologically associating domains (TADs), which represent contiguous regions with a higher frequency of intra-interactions as opposed to inter-interactions. TADs contribute to gene expression regulation by restricting the interactions between their regulatory elements, and TAD disruption has been associated with cancer. Here, we provide a proof of principle that mutations within TADs can be used to predict the survival of cancer patients. Specifically, we constructed a set of 1467 consensus TADs representing the three-dimensional organization of the human genome and used Cox regression analysis to identify a total of 35 prognostic TADs in different cancer types. Interestingly, only 46% of the 35 prognostic TADs comprised genes with known clinical relevance. Moreover, in the vast majority of such cases, the prognostic value of the TAD was not directly related to the presence/absence of mutations in the gene(s), emphasizing the importance of regulatory mutations. In addition, we found that 34% of the prognostic TADs show strong structural perturbations in the cancer genome, consistent with the widespread, global epigenetic dysregulation often observed in cancer patients. In summary, this study elucidates the mechanisms through which non-coding variants may influence cancer progression and opens new avenues for personalized medicine.

## 1. Introduction

The need for an efficient high-level organization of the genomic DNA, especially in a spatial sense, is obvious given the enormous amount of information stored in the eukaryotic genome. Thus, the human genome is three-dimensionally organized into topologically associated domains (TADs), which are hundreds of kilobases to megabases (Mb) in size and encompass multiple genes and regulatory elements [1]. TADs were originally defined based on Hi-C interaction matrices as genomic blocks exhibiting preferential physical interactions within them in contrast to between them [2]. Binding sites for the insulator protein CCCTC-binding factor (CTCF) and the protein complex cohesin were later found to be enriched in TAD boundary regions (TBRs) and to contribute to the confinement of chromatin loops within the TADs [3]. This organization into domains and loops is thought to serve as a foundation for the interaction of regulatory elements, which in turn mediates gene expression, e.g., between promoters and enhancers. Supporting this assumption, TADs have been shown to play a crucial role in gene regulation during evolution and development [4,5]. Comparative analysis of Hi-C data from different species and cell types has demonstrated that TADs exhibit a high level of conservation across cell types, and even across species [2,6,7,8]. However, TADs can be subject to changes, especially in the context of disease and cancer [9,10].

Large-scale structural variants (SVs) such as (structural) copy number variants (CNVs) have been reported to drive tumorigenesis by changing the number of copies of entire genes, most commonly through dosage effects [11]. Thus, many frequently mutated genes serve as effective biomarkers for cancer diagnosis, prognosis, and clinical management [12,13]. CNVs in non-coding regions can also be pathogenic, for example, by altering the number of gene regulatory elements or affecting their interactions [14,15]. Furthermore, CNVs can disrupt TADs [16,17], causing disease. While the fusion of adjacent TADs may allow enhancers from neighboring TADs to activate oncogenes (“enhancer hijacking”), the fragmentation of a TAD into sub-domains may insulate promoters and enhancers and prevent their interactions, resulting in gene dysregulation. For instance, *TAL1* and *PDGFRA*, two oncogenes associated with lymphoblastic leukemia and gliomas, respectively, have been reported to be activated through the fusion of adjacent TADs (e.g., [18,19]). Also, in prostate cancer cells, SVs were shown to fragment a TAD containing the *TP53* tumor suppressor gene into two smaller TADs, which successively resulted in the dysregulation of several genes [20]. 

Currently, clinical genomics analyses focus on protein-coding regions of the genome. Despite the progress achieved by international efforts such as ENCODE and FANTOM in the last decade, the interpretation of non-coding variants—and hence, their clinical application—remains challenging. Novel paradigms are needed to realize the full potential of genomic information in healthcare. Here, we show how the presence or absence of CNVs in TADs can be used to predict and explain patient outcome. To this end, we utilized data from “The Cancer Genome Atlas” (TCGA; [21]) to identify TADs enriched for cancer-related CNVs. Based on these TADs, we successfully modeled survival in 19 out of 25 cancer types using least absolute shrinkage and selection operator (LASSO) Cox regression models. To obtain an insight into the mechanisms by which variants involving the TADs that were identified as prognostic may lead to cancer, we separated the prognostic TADs into two groups, depending on their conservation in the cancer genome. We found that a considerable fraction of the prognostic TADs have been disrupted in cancer. Our results have important implications for the interpretation of cancer-related non-coding mutations and could motivate new strategies for personalizing cancer medicine.

## 2. Results

### 2.1. A Total of 1467 Topologically Associating Domains (TADs) Constitute the Consensus TAD Map of the Human Genome

TADs are known to be highly conserved in different tissues [2,6,7,8]. To verify this, we examined 24 different maps of topologically associating domains (TADs) inferred for different human tissues from Hi-C data (see Section 4). The TAD maps had a median of 1676 TADs and a median TAD size of 1.12 Mb. In addition, we observed a generally high pairwise similarity between the TAD maps of most tissues (see Section 4 and Figure 1A). 

To construct a single consensus TAD map representing the most prevalent features of the three-dimensional organization of the human genome, we examined the number of original TAD maps in which a given genomic region was comprised within a TAD. Specifically, we first calculated a “conservation score” and a “boundary score” within 40-kilo base (kb) windows across the entire genome (see Section 4). While the former estimates the probability of the nucleotides in the window being comprised within a TAD, the latter estimates the probability that the window encompasses one or more TAD boundaries. Since the tissues considered cannot be considered independent, and this is expected to be reflected in the three-dimensional organization of their genomes, we computed the conservation and boundary scores as weighted averages, with weights accounting for the relatedness between the tissues and describing their relative contribution to the total transcriptome diversity (see Section 4). Finally, we merged adjacent or overlapping windows as long as (1) their conservation score was greater than or equal to 0.5, and (2) their boundary score was lower than 0.5 (see Figure 1B and Section 4). With this approach, we inferred 1467 consensus TADs (see Appendix A). 

The genomic regions between the TADs are traditionally separated into either topological boundary regions (TBRs) or disorganized chromatin regions, depending on their size, with the former being smaller than the latter ([2]; see Section 4). The consensus TADs had a median size of 1.48 Mb and were separated by 1368 TBRs and 76 disorganized chromatin regions (with median sizes of ~80 kb and ~600 kb, respectively). The formation and stability of TBRs has been associated with the presence of binding sites for CCCTC-binding factors (CTCF) and transcription start sites (TSS) of housekeeping (HK) genes [2]. Indeed, we observed that binding sites for CTCF and TSSs of HK genes were enriched at TBRs compared to TADs (see Figure 1C,D and Section 4), supporting our definition of consensus TADs. 

### 2.2. Six Percent of Consensus TADs Are Enriched or Depleted for Cancer-Related CNVs

Genomic alterations such as copy number variations (CNVs) are common in cancer. To investigate patterns of recurrent mutation inside the consensus TADs, we utilized CNV data from primary tumor samples of 32 different cancer types and 10,435 patients from the TCGA Data Portal (see Section 4 and Appendix A). For the purpose of reducing potential sources of bias, we excluded very small (<1 kb) or large (>10 Mb) CNVs. Depending on the cancer type, the number of patient samples varied from 36 (CHOL, cholangiocarcinoma) to 1097 (BRCA, breast invasive carcinoma). For comparability, we further restricted the analysis to the 25 cancer types for which CNV data for at least 100 patient samples were available, and randomly selected 100 patient samples for each cancer type. The median number of CNVs per patient in the resulting dataset ranged from 3 (THCA, thyroid carcinoma) to 280 (OV, ovarian serous cystadenocarcinoma; see Figure 2A). 

Virtually all (99.6%) CNVs in the dataset (possibly partially) overlapped with a consensus TAD, to which, for simplicity, we will further refer as TADs. Moreover, due to their extent, on average, each CNV overlapped with four TADs, with 50% of the CNVs overlapping with at most three TADs (see Appendix A). Conversely, almost all TADs (99.6%) overlapped with CNVs in the genome of at least one patient. Indeed, only five TADs were completely devoid of CNVs. Overall, on average, each TAD overlapped with CNVs in the genomes of 12% (298) of the patients, with 50% of the TADs overlapping with CNVs in the genomes of at most 12% (299) patients. Only 8% (115) of the TADs overlapped with CNVs in the genomes of 20% (500) or more patients and the largest number of patients associated with a TAD was 796 (32%). Importantly, the size of the TADs was only weakly correlated with the number of overlapping CNVs (Spearman’s correlation coefficient = 0.32; see Appendix A), suggesting that TADs overlapping with very few or very many CNVs might be involved in general cancer mechanisms. 

To identify TADs with unusually large numbers of CNVs, we compared the number of patients with CNVs overlapping with each TAD, with the expectation based on its size (see Section 4). Independently of the cancer type, we found that 79 (6%) TADs were significantly enriched, with fold-differences of up to 3; in addition, the aforementioned five TADs overlapping with no CNVs were also significantly depleted, with their expectation ranging from 230 to 295 patients (see Section 4 and Figure 2B). Compared to other TADs, significantly enriched TADs were mainly associated with type I interferon response and natural killer cell activation (see Appendix A), which are known to be crucial for efficient tumor immune surveillance (e.g., [22]); depleted TADs were linked to transcriptional regulation. In addition, TADs enriched for CNV tended to be located towards the ends of the chromosomes, reflecting the genomic distribution of CNVs [23]. As expected, different cancer types showed distinct patterns of enrichment, with a median of 64 TADs being significantly enriched for CNVs. In total, 487 (33%) TADs were significantly enriched for CNVs in the genome of the patients of at least one cancer type, including the 79 aforementioned TADs. We observed only one significantly depleted TAD, only for skin cutaneous melanoma (SKTM). Most (303) of the significantly enriched TADs were enriched for CNVs in the genome of the patients of two or more patient types (see Figure 2C). Moreover, although no TAD was significantly enriched for CNVs in the genome of the patients of all cancer types, 13 TADs were for 15 or more cancer types and might be considered the basis of a pan-cancer mutational signature. Intriguingly, these 13 TADs were not enriched for genes with common alterations across different cancer types (“pan-cancer genes”; see Section 4), indicating that other mechanisms are likely to be involved.

### 2.3. TADs Enriched for CNVs Are Valuable Prognostic Biomarkers in Cancer

In order to assess the value of TADs enriched for CNVs in cancer prognosis, we trained LASSO Cox regression models for each of the 19 out of 25 cancer types for which at least 100 patients were available and for which at least 10% of the patients had a lethal outcome (see Section 4). The models were based on age, sex, and the presence/absence of CNVs in the TADs enriched for CNV in each particular patient cohort. The models for glioma tumors (GBM and LGG), clear cell and papillary renal carcinomas (KIRC and KIRP), gynecologic carcinomas (OV, BRCA, and UCEC), rectum adenocarcinoma (READ), and sarcoma (SARC) displayed median c-indices between 0.55 and 0.8 and were therefore considered reliable. In order to establish a reference for evaluating the predictive power of our TAD-based models, we compared them to analogous nineteen gene-based models, trained and tested in the exact same manner (see Section 4). We found that only six of the gene-based models were considered reliable, and that all the corresponding TAD-based models were considered reliable as well (see Figure 3A). Moreover, the TAD-based models performed better than the gene-based models in four cancer types (BRCA, OV, SARC, UCEC; *p*-values < 0.05, Wilcoxon’s rank-sum test), worse in three (KIRP, KIRC and GBM; *p*-values < 0.05, Wilcoxon’s rank-sum test), and similarly in two (LGG and READ). 

The prognostic features of the TAD-based models can be used to stratify patients into high- and low-risk groups (all *p*-values <0.001, log-rank test; see Section 4, Figure 3B–E, and Appendix A). In addition to increasing age, which was generally associated with lower survival (6/9 models), we obtained a total of 35 prognostic TADs (see Section 4 and Figure 3F). Only three of these TADs were shared by at least two different cancer types (chr9:21240000-24400999 by LGG, GBM, and KIRC; chr20:57399000-58159999 by BRCA and OV; and chr12:55679000-57720999 by GBM and KIRP), suggesting that the identity of the TADs harboring structural variants is mainly cancer-specific. With few exceptions (5 out of the 35 prognostic TADs; see Appendix A), we observed no differences between copy number gains and losses, although the power to detect such effects is limited by the small sample sizes.

Interestingly, most (19, 54%) prognostic TADs did not comprise any pan-cancer genes (see Section 4 and Figure 3F). Consequently, in most cases, the presence of pan-cancer genes does not explain the predictive power of the TADs. A remarkable case was SARC, because only the TAD-based models exhibited a reliable performance and because none of the five TADs that were prognostic for SARC comprised any pan-cancer genes (see Figure 4A,B and Appendix A). Furthermore, among the 16 TADs that comprised at least one pan-cancer gene (four in KIRC, three in LGG, four in GBM, one in UCEC, three in KIRP, one in BRCA, two in OV, and one in READ, with one of these prognostic TADs being shared by GBM, LGG, and KIRC and another one by KIRP and GBM; see Figure 3F), the prognostic effect of the TAD was not necessarily directly related to the pan-cancer gene(s). In fact, seven of these TADs did not comprise any prognostic pan-cancer genes, and among the remaining nine, only two TADs showed lower prediction power than the prognostic pan-cancer genes that they comprised (chr7:54760000-58079999, a prognostic TAD in LGG, comprising the prognostic pan-cancer gene EGFR; and chr12:55679000-57720999, a prognostic TAD in GBM, comprising the prognostic pan-cancer gene DDIT3; see Section 4, Figure 4C,D, and Appendix A). In other words, in most cases, the survival of the patients with CNVs affecting the pan-cancer gene(s) in a TAD did not differ to that of the patients with CNVs in the TAD that did not affect the pan-cancer gene (see Section 4) and the presence of a CNV in the pan-cancer gene was as informative as the presence of a CNV within the rest of the TAD. This has enormous implications for patient stratification for prognosis. For example, the TAD chr9:21240000-24400999, which is prognostic for LGG, comprises the prognostic pan-cancer gene *CDKN2A*. LGG patients with CNVs in this TAD (*n* = 96) exhibited lower survival than patients with no CNVs in this TAD (*n* = 414; *p* < 0.0001, log-rank test), and at least part of the predictive power of the TADs was not associated with the gene: Namely, the survival of the patients with CNVs affecting *CDKN2A* (*n* = 87) did not differ from that of the patients with CNVs in the TAD that did not affect *CDKN2A* (*n* = 9; log-rank test; see Figure 4E,F). This shows that the mutations involving the sequence of *CDKN2A* are as relevant to LGG prognosis as those that do not, and that the latter should not be disregarded. 

These results show that a large proportion of prognostic TADs exhibit predictive power independent of pan-cancer genes and illustrate the potential of TAD-based models to complement traditional gene-based models.

### 2.4. Thirty-Four Percent of Prognostic TADs Tend to Undergo Large Structural Changes in Cancer

To gain further biological insight into the prognostic features of the TAD-based models, we constructed a consensus TAD map representing the most prevalent features of the three-dimensional organization of the human cancer genome (see Section 4). This cancer TAD map comprised 1622 (consensus) TADs. When comparing the TAD map constructed for the “normal”, healthy human genome to the cancer TAD map, we found that 44% (643, covering 37% of the genome) of the TADs in the former had a highly similar counterpart in the latter (i.e., with a reciprocal overlap ≥95%; see Section 4) and can be considered “constitutive”, 34% (505, covering 29% of the genome) showed marked differences (i.e., they overlapped with less than 70% of any cancer TAD and/or less than 70% of the normal TAD overlapped with any cancer TAD; see Section 4 and Appendix A) and are “perturbed”, and 22% (319) displayed intermediate similarities and are, thus, ambiguous with regards to their conservation between the “normal” and cancer states (see Appendix A). These results are in agreement with the expected high degree of conservation of the TADs across tissues/cell lines, but also bring to light differences that may be associated with altered regulatory interactions in cancer.

In general, constitutive and perturbed TADs did not display any differential enrichment for CNVs. Nevertheless, the distribution of CNVs along the TADs exhibited a clear trend, with CNVs being enriched at the center of the TADs as compared to their boundaries and immediately adjacent TBRs (see Figure 5A and Section 4). These findings support an association between TAD boundaries and CNVs. We hypothesize that CNVs disrupt the TAD boundaries and that because TAD disruption can rewire entire gene expression programs, it is very likely deleterious. This would explain the pressure to preserve TAD boundaries. Moreover, we observed that two subsets of perturbed TADs outstandingly deviated from the rest: A set of 120 TADs that are split into two or more TADs in the cancer genome and a set of 111 TADs that are fused with one or more other TADs in the cancer genome. While the former showed a stronger enrichment towards the center, the latter featured generally high enrichment, independently of the relative position in the TAD (see Figure 5A and Section 4). Indeed, under our assumption, TADs that are split in the cancer genome would be expected to be especially enriched for CNVs towards the body of the TAD, away from their boundaries, while TADs that are fused in the cancer genome would show enrichment for CNVs at the boundaries. Thus, whereas the CNVs would induce the formation of new boundaries in the former, they would lead to the elimination of existing boundaries in the latter.

While 43% of the 35 prognostic TADs were constitutive, 34% were perturbed, indicating that at least one-third of the predictive power of the TADs could be directly associated with changes in the three-dimensional organization of the genome. The ratio of constitutive to perturbed prognostic TADs was similar for different cancer types. For instance, out of the eight TADs that are prognostic for LGG—the best performing models— three were constitutive and four perturbed. Also, out of the five TADs that are prognostic for SARC—for which only the TAD-based models performed reliably— three were constitutive and two perturbed. Many of the perturbed prognostic TADs appear to have been transformed through multiple splitting and fusion events in the cancer genome and were associated with local changes in the number of patients with CNVs. Thus, chr9:21240000-24400999 and chr9:24431000-24564999, two of the perturbed prognostic TADs for LGG, showed a relatively high number of CNVs, with a peak at the location of the pan-cancer gene *CDKN2A* (see Figure 5B); in the region comprising the perturbed prognostic TAD for LGG chr1:8000000-10440999, the number of CNVs decreased towards the 3′ end (see Figure 5C); and for the remaining perturbed prognostic TAD for LGG, chr3:196040000-198159999, we observed a general increase in the number of CNVs (see Figure 5D). Similarly, while the perturbed prognostic TAD for SARC chr17:70680000-73360999 was split into two smaller TADs in the cancer genome (see Figure 5E), the remaining perturbed prognostic TAD for SARC, chr12:125039000-128113999, was fused with a neighboring TAD (see Figure 5F); both TADs were linked to a local increase in the number of CNVs. Remarkably, chr12:125039000-128113999 only comprises two protein-coding genes: *TMEM132B* and *AACS.* None of these genes have been reported to be associated with SARC. Its adjacent TAD chr12:123760000-124919999—with which it is fused in the cancer genome (see Figure 5F)—comprises, among other genes, the pan-cancer gene *NCOR2*. *NCOR2* is a prognostic gene for SARC. Consistently, SARC patients with CNVs in *NCOR2* (*n* = 84) exhibited significantly lower survival than patients without CNVs in *NCOR2* (*n* = 174, *p* < 0.01, log-rank test). Also, the patients with CNVs in these genes featured higher *NCOR2* expression levels (median FPKM values 6122 vs. 5035, *p* < 0.01, Wilcoxon rank-sum test; see Section 4). Interestingly, the survival of the patients with CNVs in *NCOR2* did not differ from that of the patients with no CNVs in *NCOR2,* but with CNVs in chr12:125039000-128113999 (*n* = 30). Moreover, the two groups of patients did not show any differences in their *NCOR2* expression levels (see Appendix A). In this context, it is reasonable to hypothesize that the fusion of chr12:125039000-128113999 and chr12:123760000-124919999 leads to enhancers in chr12:125039000-128113999—which, under normal conditions, would not interact with *NCOR2*—being hijacked by *NCOR2*. Aberrant expression of *NCOR2* has been associated with several cancers [24]; in particular, the complex formed by *NCOR1* and *NCOR2* in concert with *HDAC3* epigenetically suppresses myogenic differentiation in Embryonal rhabdomyosarcoma (ERMS), which is required for tumor growth [25].

In summary, except for *CDKN2A*, which is a clear outlier in the dataset under consideration, the changes in the local number of CNVs did not appear to be connected to the pan-cancer genes located in these regions (*CAMTA1* and *MTOR*, *MUC4* and *TFRC*, and *NCOR2*; see Figure 5C,D,F, respectively). Our findings exemplify how comparing between the three-dimensional structure of the healthy and cancer human genomes can provide insights into the mechanisms by which non-coding variants lead to gene expression dysregulation in cancer.

## 3. Discussion

TADs are known to be generally well conserved across cell types and species [2,6,7,8]. To obtain a consensus TAD map representing the most prominent three-dimensional features of the human genome, we integrated the information from different TAD maps generated using Hi-C on 24 normal tissues. These tissues represent a wide coverage of the human body and were selected based on availability of Hi-C data and corresponding TAD maps. Importantly, the Hi-C data for this collection of samples were generated by a relatively small number of laboratories using similar protocols, and processed with the exact same computational pipeline [2]. Indeed, the Hi-C protocol is complex, and variations at the numerous steps can affect the resulting data [26,27]. Pairwise comparisons of the TAD maps from the 24 normal tissues indicated only minor batch effects associated with the laboratory that generated the data. Integrating further TAD maps would imply being able to distinguish batch effects from biological variability, which is not possible with the current understanding of the factors that contribute to the success of a Hi-C experiment. Moreover, the definition of our consensus TAD maps depends on several parameters. Specifically, we used a sliding window approach to decide whether each 40-kb-long genomic window is part of a consensus TAD or not based on the number and type of tissues in which it was found to be part of one of the original TADs. This is controlled by two parameters: The conservation score and the boundary score. We empirically set both parameters to 0.5; effectively, this means that, on average, the genomic window of interest is part of a TAD in 50% of the TAD maps. Due to the high similarity observed between the TAD maps of different tissues, our consensus TADs are not expected to change dramatically upon moderately increasing this threshold. Finally, the approach takes into account that closely related tissues—as assessed from their gene expression profiles—are expected to show similar TAD maps, accounting for possible biases in the collection of samples. It will be interesting to revisit these analyses as broader Hi-C datasets become available. 

Many types and locations of CNVs have been linked to cancer. To look for associations between the presence/absence of CNVs in specific TADs and cancer prognosis, we utilized CNV data from TCGA. TCGA is a rich resource comprising different genomic and transcriptomic data types for hundreds of patients from 32 different cancer types. There are certainly other similar resources, such as the data portal of the International Cancer Genome Consortium (ICGC) [28], which we could have employed in the analysis. Although the integration of such datasets would have resulted in larger cohorts and higher statistical power, their integration is not trivial and would have increased the number of unwanted artifacts. Using the TCGA dataset, we identified a group of 79 TADs susceptible to undergo copy number changes compared to random genomic sequences of the same size, and found that the specific TADs varied with the cancer type. Together with TBRs and disorganized chromatin regions, the TADs form a partition of the genome. In particular, the TADs account for ~85% of the genome. Nevertheless, the aforementioned 79 TADs occupy only ~6% of the genome, suggesting that only a small fraction of the genome is subjected to changes in multiple patients and could be used to model overall survival. Using LASSO Cox regression modeling, we showed that in 47% (9/19) of the assessed cancer types, the TADs enriched for CNVs were informative for patient prognosis. Furthermore, the TAD-based models performed similarly to models trained on 717 pan-cancer genes. Given the limited data available, we used a cross-validation framework to reduce the risk of overfitting our models. The performance of our OV model on an independent patient cohort (c-index = 0.59; see Section 4 and Appendix A) suggests that our models—at least, the one for OV—are able to generalize successfully to new data. The TAD-based models for sarcoma (SARC), which achieved a median c-index of 0.58, is a notable case. SARC patients usually harbor a relatively large number of variants in non-coding regions and no very good gene markers are known [29,30,31]. Indeed, our gene-based models were unable to reliably predict survival (median c-index = 0.52). In contrast, using our TAD-based models identified five prognostic TADs, none of which comprised any pan-cancer genes. This observation illustrates the value of the TADs as prognostic biomarkers and their potential to improve upon gene-based models. 

Remarkably, our TAD-based models provide insights into the mechanisms by which non-coding variants may contribute to cancer progression. In total, we identified 35 prognostic TADs for nine different cancer types. Although functional assays are necessary to validate our findings, a relatively large fraction of these TADs appear to be associated with cancer-related changes in the three-dimensional organization of the genome. Indeed, among the 35 prognostic TADs, 34% were perturbed and, hence, had undergone changes in the majority of the cancer types examined. In addition, some constitutive TADs may actually be perturbed in a cancer type-specific manner. It is worth noticing that the distinction between constitutive and perturbed TADs depended on arbitrary thresholds on the overlaps between the normal and cancer TAD maps. The chosen thresholds were relatively strict; in particular, the threshold set for the constitutive TADs (95% reciprocal overlap) was chosen to reflect the expected high similarity, but also to allow for smaller rearrangements or variations in the protocols used for generating the underlying Hi-C data; and the perturbed TADs may display small structural alteration considering the high stability of TADs. Analogously, the threshold set for perturbed TADs (at most 70% for one or both of the two overlaps considered; see Section 4) was set to enable the identification of TADs with relatively large differences. Thus, the 22% of ambiguous TADs can be viewed as a safety buffer, since these represent TADs that cannot be unequivocally classified as either constitutive or perturbed. Changing the thresholds has a foreseeable effect on the classification; for example, while the number of constitutive TADs increases from 643 to 887 if we require a lower reciprocal overlap (80%), the number of perturbed TADs decreases from 505 to 274 if we enforce a lower maximum overlap (50%).

Despite the fact that the TADs are expected to be conserved between individuals, the comparison of the DNA-seq and Hi-C data of the same patient would enable a more direct examination of the effects of CNVs on the TADs. Moreover, successfully modeling certain cancer types in specific human populations may require Hi-C data representing the peculiarities of the three-dimensional structure of their genomes. However, to date, TCGA does not contain Hi-C data. In fact, only a small number of patient Hi-C datasets have been generated. For instance, Kloetgen et al. have produced and studied Hi-C data for six primary T-cell acute lymphoblastic leukemia (T-ALL) patients [32], and Díaz et al. have done so for one large B-cell lymphoma patient [33]. To conduct an independent validation of some of our results, we compared a list of 37 TADs that have been reported to undergo structural changes in the B-cells of lymphoma patients [33] to our normal consensus TADs; we observed that 78% (29/37) of these 37 TADs overlapped by at least 90% with our consensus TADs, but only 8% (3/37) overlapped with constitutive TADs, while the expectation based on their sizes is 27% (10/37). This observation supports our consensus TAD map and its utility to represent the normal, healthy three-dimensional structure of the genome.

Finally, we would like to stress that the performances of both the TAD- and gene-based models could be optimized, for example, by integrating other kinds of data—e.g., clinical and RNA-seq data—into the prediction or by opting for another feature section method. Our aim was simply to carry out a proof of principle study to demonstrate the value of TADs as prognostic markers for cancer. Although some caution is warranted until our findings are externally replicated, both our internal validation and the existing literature support the predictive power of TAD-based models.

## 4. Materials and Methods

### 4.1. Topologically Associating Domain (TAD) Maps

TAD maps for the human genome (hg38) were downloaded from http://promoter.bx.psu.edu/hi-c/publications.html (accessed on 8 August 2018; Appendix A). The underlying Hi-C data were generated in the course of six different studies [2,7,34,35,36], but processed with the same bioinformatics pipeline [2]. These TAD maps represent the three-dimensional organization of the genome in 24 human healthy (“normal”) tissues/cell lines and 11 cancer cell lines. Note that the normal and cancer samples are not completely separable from each other, which may, to some extent, be explained by the relatedness of the tissues/cell lines. Nevertheless, 58% of the cancer samples appeared to be more related to each other than to most (92%) normal samples (see Appendix A).

### 4.2. TAD Size Comparison Between Normal and Cancer States 

We computed the median size of the TADs identified in each tissue/cell line, and then the median of the medians across (1) normal tissues/cell lines, and (2) cancer cell lines.

### 4.3. Similarity Between TAD Maps of Different Tissues/Cell Lines

Given a pair of TAD maps A and B, we searched for TADs in B overlapping each TAD in A; if one or more overlaps were found in B for a TAD in A, we recorded the TAD covering the largest fraction of the TAD in A; if no overlaps were found, we recorded that as 0. The median across all TADs in A was used to represent the similarity between A and B. Note that this definition of similarity is not symmetric in A and B. 

### 4.4. Construction of Consensus TADs

#### 4.4.1. Contribution of Each Tissue/Cell Line in Total Gene Expression Divergence

We obtained total RNA-seq read counts from the ENCODE data repository (https://www.encodepsroject.org/, accessed on 28 August 2018; [37]) for each of the tissues/cell lines corresponding to the TAD maps of interest. For those tissues/cell lines for which total RNA-seq data were not available, we used poly-A RNA-seq (see Appendix A). We combined the data for all protein-coding genes reported by the ENCODE project (https://www.encodeproject.org/search/?type=Experiment&status=released&assay_title=total+RNA-seq&award.project=ENCODE&assembly=GRCh38, accessed on 29 August 2019; [37]) and transformed the count data using the vst() function with parameter “blind=TRUE” (for a fully tissue/cell line unaware transformation) in the DESeq2 R/Bioconductor package (version 1.16.1; [38]). We next ranked the genes according to their maximum absolute deviation from the median expression value across all tissues/cell lines and selected the 2000 genes with the highest ranks. Then, we computed the Pearson correlation coefficient between the expression profiles of these 2000 genes of all pairs of tissues/cell lines and hierarchically clustered the tissues/cell lines based using average linkage. The resulting dendrogram represents the relationships among the tissues/cell lines based on their gene expression profiles (see Appendix A). We then applied BranchManager (BM; [39]) to compute a “weight” that summarizes the contribution of each tissue/cell line to the total gene expression divergence according to the topology and branch length of the dendrogram. BM assumes that the differentiation of the tissues/cell lines is a Brownian process in which each tissue/cell line can be regarded as an endpoint. Based on this, it infers the differentiation trajectories of the tissues/cell lines and each tissue/cell line is assigned a weight equal to its contribution to the inference. Specifically, BM computes the trajectories from the centroid to all the tissues/cell lines in the N-dimensional space in which the dendrogram can be embedded, where N is the number of tissues/cell lines in the analysis. 

#### 4.4.2. Conservation Scores

Each TAD map was assigned one of the aforementioned weights according to the tissue/cell line from which the underlying Hi-C data originated. Two different TAD maps were available for the cell lines GM12878 and K562; in both cases, each of two TAD maps was assigned half of the weight computed for the corresponding cell line. The TAD maps and their weights were used to compute a consensus TAD map representing the majority of the tissues/cell lines considered. Specifically, we computed a TAD “conservation” score and a TAD “boundary” score for 40-kb-long sliding windows across the entire genome. Given n TAD maps, we defined the conservation score of the $i^{th}$ window of length L (here, L=40000) as:
(1)$$c_{i}=\frac{1}{L}\overset{L+i-1}{\underset{j=i}{\sum }}\overset{n}{\underset{k=1}{\sum }}w_{k}\cdot I_{T}\left(j,k\right)$$
where $w_{k}$ is the weight for the $k^{th}$ TAD map and
(2)$$\begin{array}{c} I_{T}\left(j,k\right)\\ =\left\{\begin{array}{cc} 1, & \mathrm{if}\;\mathrm{the}\;j^{th}\;\mathrm{nucleotide}\;\mathrm{of}\;\mathrm{the}\;\mathrm{window}\;\mathrm{is}\;\mathrm{within}\;a\;\mathrm{TAD}\;\mathrm{in}\;\mathrm{the}\;k^{th}\;\mathrm{TAD}\;\mathrm{map}\\ 0, & \mathrm{otherwise}. \end{array}\right. \end{array}$$
and the “boundary” score as:
(3)$$b_{i}=\overset{n}{\underset{k=1}{\sum }}w_{k}\cdot I_{B}\left(i,k\right)$$
where
(4)$$I_{B}\left(i,k\right)=\left\{\begin{array}{cc} 1, & \mathrm{if}\;\mathrm{the}\;i^{th}\;\mathrm{window}\;\mathrm{includes}\;a\;\mathrm{TAD}\;\mathrm{boundary}\;\mathrm{in}\;\mathrm{the}\;k^{th}\;\mathrm{TAD}\;\mathrm{map}\\ 0, & \mathrm{otherwise}. \end{array}\right.$$

The windows were shifted by 1000 base pairs (bp); i.e., 97.5% overlap between adjacent sliding windows. The TAD conservation and boundary scores ranged from zero to one.

Finally, we merged all adjacent windows as long as (1) the nucleotide-wise average TAD conservation score was ≥0.5, and (2) none of the windows had a TAD boundary score ≥0.5. The resulting genomic regions were defined as consensus TADs if their size was ≥40 kb. A region between two adjacent consensus TADs was considered as a TBR if its length was ≤400 kb, and as an “unorganized chromatin region” otherwise (as in [2]).

### 4.5. Constitutive and Perturbed TADs

We divided the normal consensus TADs into three groups, according to both the fraction of their sequence overlapping with a cancer TAD and the fraction of the sequence of the corresponding cancer TAD overlapping with it. Specifically, for each normal TAD $N_{i},$ we recorded the fraction of its sequence overlapping with a cancer TAD. If it overlapped with two or more cancer TADs, we recorded the highest fraction. In turn, we examined the corresponding cancer TAD $C_{i}$, and recorded the fraction of its sequence overlapping with a normal TAD. Analogously, if the cancer TAD overlapped with two or more normal TADs, we recorded the highest fraction. Let the corresponding normal TAD be $N_{j}$. If $N_{i}=N_{j}$ we called the relationship between $N_{i}$ and $C_{i}$ “reciprocal”. Then, we defined (a) (normal) TADs with reciprocal fractions ≥0.95 as “constitutive” TADs; (b) (normal) TADs with fractions ≤0.7 and/or for which the corresponding cancer TAD had a fraction ≤ 0.7 as “perturbed” TADs; and (c) (normal) TADs that did not satisfy any of the above criteria as “ambiguous” TADs. 

### 4.6. Enrichment Analysis of CTCF Peaks and Housekeeping Genes (HK Genes) in Consensus TADs and TBRs

CTCF ChIP-seq datasets matching 17 of the normal and cancer tissues and cell lines used to construct the consensus TADs were obtained from the ENCODE data repository (https://www.encodeproject.org/, accessed on 28 August 2018; [37]). A total of 6290 housekeeping genes were downloaded from https://www.tau.ac.il/~elieis/HKG/ (accessed on 28 August 2018; [40]). CTCF and housekeeping gene densities were computed with the “reference-point” sub-command of the computeMatrix command from deepTools (version 3.1.2; [41]) with parameters --referencePoint center -a 200000 -b 200000 --binSize 5000 --missingDataAsZero. The plotProfile command was used for visualization.

### 4.7. Cancer-Related Copy Number Variants (CNVs)

We collected masked somatic copy number variation (CNV) for the 25 different types of cancer with at least 100 primary tumor samples in TCGA database (see Appendix A). We only considered the CNVs longer than 1 kb and up to 10 Mb with a segment mean larger than 0.1 or smaller than −0.1, and a number of probes of at least 10. 

### 4.8. TADs Enriched/Depleted for CNVs

For each consensus TAD, we selected a random region from the human genome with the same size. The procedure was repeated 1000 times. We then computed the number of patients exhibiting CNVs in that particular TAD and compared it to those computed for its random counterparts. A patient was considered to exhibit a CNV in a TAD if the CNV overlapped with the TAD by at least 1 bp. A TAD was considered enriched for CNVs if, at most, 5% of its random counterparts displayed a number of patients greater than or equal to that observed for the TAD. A TAD was considered depleted for CNVs if, at most, 5% of its random counterparts displayed a number of patients smaller than or equal to that observed for the TAD.

### 4.9. Pan-Cancer Genes

A total of 717 pan-cancer genes were downloaded from the COSMIC database (“Cancer Gene Census” file; release v89, https://cancer.sanger.ac.uk/cosmic, last accessed on 15 May 2019; [42]). The TADs were defined as comprising pan-cancer genes if the pan-cancer genes overlapped with TADs with at least 1 bp. 

### 4.10. Functional and Pathway Analysis

Functional and pathway analysis was performed with the Database for Annotation and Integrated Discovery (DAVID Bioinformatics Resources 6.8, http://david.abcc.ncifcrf.gov/; [43,44]). In particular, we focused on the ontologies KEGG_PATHWAY, Biological Processes (BP), Molecular Function (MF), and Cellular Component (CC). Terms associated with a false discovery rate (FDR) ≤0.05 were considered significantly enriched. 

### 4.11. Survival Analysis

Overall survival was analyzed using Cox regression, Kaplan–Meier curves, and log-rank tests.

#### 4.11.1. TAD- and Pan-Cancer Gene-Based Overall Survival Cox Regression Models

LASSO Cox regression analysis [45] was used to construct TAD- or gene-based models for predicting overall patient survival and to identify prognostic TADs/pan-cancer genes. In particular, we trained and tested models for the 19 cancer types in the dataset that were represented by at least 100 patients and had at least 10% of lethal outcomes using TADs/pan-cancer genes, age, and sex as predictors or “features”. With the exception of those patients with no known CNVs in their genomes, all the available patients were included in this analysis. Overall patient survival was defined as the time to death or to the last follow-up date of the patient. The analysis was conducted with functions of the glmnet R package [46]. 

The patient samples of each cancer type were randomly separated into a training (2/3) and a test (1/3) set. We then identified TADs significantly enriched for CNVs in the genomes of the patients in the training set and recorded the presence (encoded as 1) or absence (encoded as 0) of CNVs for each of those patients in the identified TADs. In addition, we encoded the age of the patient as 1 if it was larger than the median age in the training set, and 0 otherwise; and the sex of the patient as 1 for females, and 0 for males. Finally, we performed a LASSO Cox regression analysis on the TADs enriched for CNVs, age, and sex. To choose the value of the LASSO regularization parameter (λ), we performed a 5-fold cross-validation using the cv.glmnet() function with parameters ‘family = “cox”, alpha = 1, nfold = 5′ and computed the value of λ leading to the minimum prediction error (“lambda.min”). We then extracted the coefficients of the model trained with the glmnet() function with parameters ‘family = “cox”, alpha = 1′ for this particular value of λ, sorted them based on their absolute values in decreasing order, and selected the top ten. In cases in which cv.glmnet() failed to converge (the maximum number of iterations was the default, 100,000), we selected the top ten features based on the glmnet() ranking according to their appearances in the regularization path. The performance of the model implicitly defined by these coefficients was evaluated on the test set using the c-index [47]. The entire procedure above was repeated 1000 times. Thus, for each cancer type, we trained and tested 1000 models.

#### 4.11.2. Prognostic TADs and Pan-Cancer Genes

To select the features to be included in the final model, which was trained on all available patients, we applied an approach similar to that of Laimighofer et al. [48]. First, for the $i^{th}$ model we computed a weight:
(5)$$w_{i}=\left\{\begin{array}{c} \begin{array}{cc} \frac{1}{K}\cdot \exp\left(\log\;\left(2\right)\frac{devCI_{i}}{0.1}\right), & \mathrm{if}\;CI_{i}\geq 0.5 \end{array}\\ \begin{array}{cc} 0, & ifCI_{i}<0.5 \end{array} \end{array}\right.$$
where K is the total number of models (here, K=1000) and $devCI_{i}=CI_{i}-\frac{1}{K}\sum_{j=1}^{K}CI_{j}$. Next, we normalized the weights $w_{i}$ to add up to one:
(6)$$w_{i}^{\prime }=\frac{w_{i}}{\sum_{j=1}^{K}w_{j}}$$

Then, we defined the aggregated normalized weight of the $j^{th}$ feature as:
(7)$$P_{j}=\overset{k}{\underset{i=1}{\sum }}w\prime_{i}\cdot I\left(j,i\right)$$
where $I\left(j,i\right)=1$ if the $j^{th}$ feature was selected in the $i^{th}$ model, and $I\left(j,i\right)=0$ otherwise.

The ten features with the highest aggregated normalized weights were selected to train the final Cox regression model. In turn, the features of the final model were selected using backward stepwise elimination. Features with regression coefficients significantly (*p* ≤ 0.05) different from zero (assessed through separate Wald tests) were considered prognostic. The *p*-values of the features in the model were corrected for multiple testing using the FDR.

#### 4.11.3. Patient Stratification

The hazard ratios derived from the final Cox regression model were used to calculate a “risk score” for each patient. The risk score was defined as the sum of the weighted values for the prognostic features, with the weights being the coefficients estimated by the Cox regression model. The patients were separated into high- and low-risk groups according to the median risk score across all patients. 

Kaplan–Meier (KM) survival analysis was performed to compare the overall survival of different groups of patients. The log-rank test was used to determine differences between the groups; a *p*-value ≤0.05 was considered significant. These analyses were performed with the ggsurvplot() function in the survival R package [49], with default parameters. 

### 4.12. CNV Densities in Constitutive and Perturbed TADs

CNV densities in all (normal) constitutive, all (normal) perturbed TADs and two subsets of (normal) perturbed TADs were computed with the “scale-regions” sub-command of the computeMatrix command from deepTools (version 3.1.2, [41]) with parameters “--averageTypeBins median -m 2000000 -a 1000000 -b 1000000 -bs 10000 --missingDataAsZero”. The plotProfile command was used for visualization. Specifically, we extracted (1) (normal) perturbed TADs that are split into multiple TADs in the cancer genome such that two or more of the corresponding cancer TADs overlap with original (normal) TAD by at least 95% of their sequence; and (2) (normal) perturbed TADs that are fused together into one TAD in the cancer genome, with two or more of such TADs overlapping by at least 95% of their sequence with the corresponding cancer TAD.

### 4.13. Expression Levels of NOCR2 in SARC Patients

Fragments Per Kilobase of transcript per Million mapped reads (FPKM) for NOCR2 for all TCGA SARC patients were downloaded from OncoLnc (http://www.oncolnc.org/, accessed on 18 September 2019; [50]). 

### 4.14. Independent Validation of the OV TAD-Based Model

An additional cohort comprising 93 independent ovarian cancer-serous cystadenocarcinoma (“OV-AU”) patients was obtained from the data portal of the ICGC (https://dcc.icgc.org/projects/OV-AU, last accessed on 14 November 2019). ICGC contains copy number variants (“Copy Number Somatic Mutations (CNSM)”, under the aforementioned link) and survival and vital status for all the patients (“Clinical Data”, under the aforementioned link). We used these data in an analogous manner, as we did with the data from TCGA, to test the final TAD-based LASSO Cox regression model that we trained on TCGA data for OV. 

## 5. Conclusions

To our knowledge, this is the first time this has been investigated in a systematic manner. Our TAD-based models capture effects of CNVs in non-coding regions and are, thus, complementary to traditional gene-based models. In particular, our data hint towards a substantial fraction of prognostic features being linked to changes in the three-dimensional organization of the human genome. More generally, this study provides a framework for prioritizing non-coding variants for the development of personalized cancer therapies. The rapid increase in the number of available patient Hi-C datasets promises to improve the efficacy of TAD-based models in cancer diagnosis and prognosis in the near future.

## Figures and Tables

**Figure 1 cancers-11-01886-f001:**
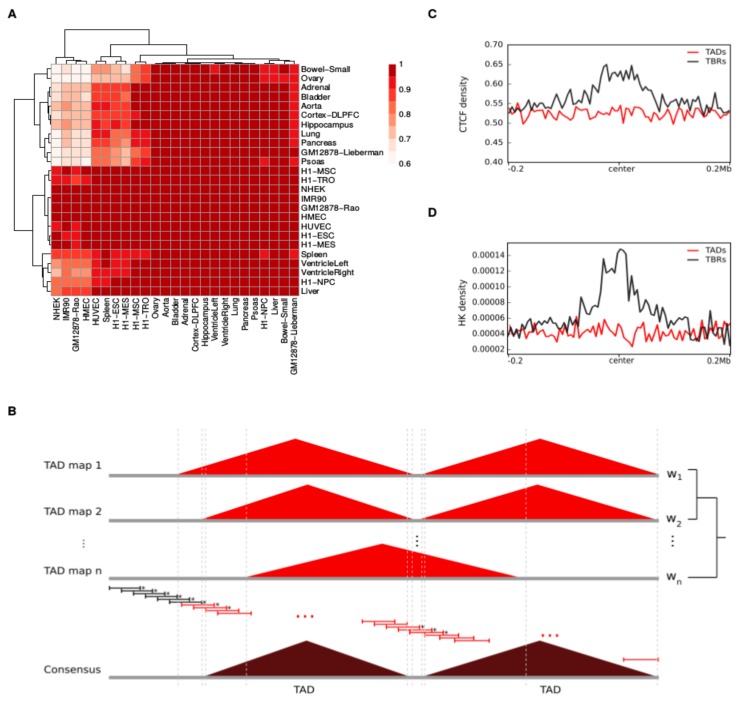
Topologically associating domain (TAD) maps across various human tissues can be combined into a consensus TAD map. (**A**) TAD maps of different human tissues generally showed high similarity. The heatmap summarizes the pairwise similarities of every pair of samples; specifically, we computed the maximum fraction of a TAD in the *j*th sample (columns) that overlapped with a TAD in the *j*th sample (rows). The color of the cells represents the median of the fractions for all TADs in the *j*th sample. Note that this is not symmetric. Rows and columns are clustered based on Euclidean distance using complete linkage. (**B**) The consensus TADs were derived by computing TAD conservation and boundary scores in sliding windows across the genome. Adjacent windows were merged as long as they had a nucleotide-wise TAD conservation score ≥0.5 and none of the windows had a TAD boundary score ≥0.5, and defined as consensus TADs if their size was ≥40 kb. (**C**,**D**) TADs and topological boundary regions (TBRs) showed distinct coverage of CCCTC-binding factors (CTCF) and housekeeping (HK) genes. Density of CTCF (**C**) and housekeeping genes (**D**) calculated for 5-kb-long bins covering the center of the TADs (red)/TBRs (black) ±200 kb.

**Figure 2 cancers-11-01886-f002:**
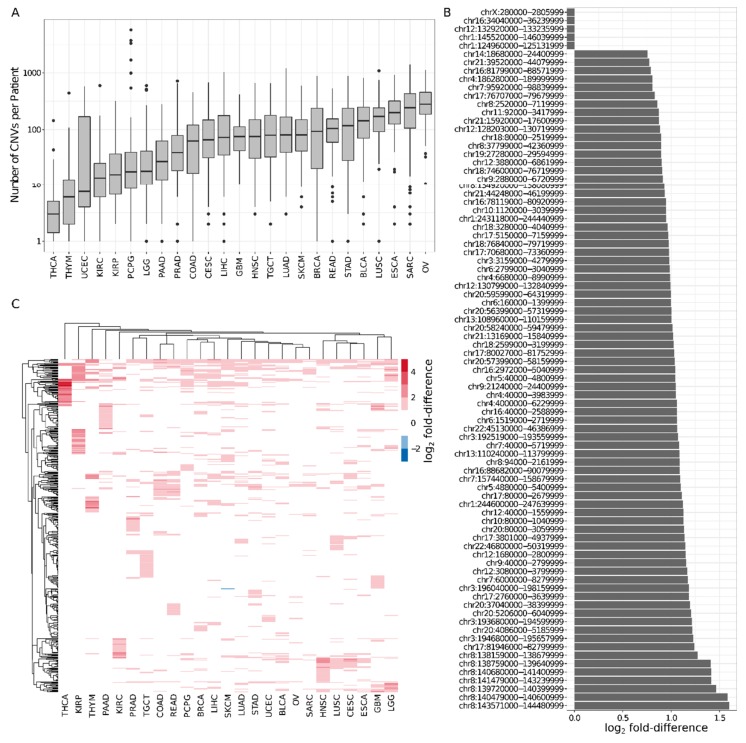
Some of the consensus TADs are enriched or depleted for cancer-related copy number variations (CNVs). (**A**) The number of CNVs per patient differed among the 25 considered cancer types. The number of CNVs per patient (y-axis) is represented on a log_10_ scale. (**B**) Seventy-nine TADs were enriched and 5 TADs were depleted for CNVs across all the patients of all 25 cancer types. The coordinates of the TAD are indicated on the y-axis; the bars show log2 fold-differences to the genomic expectation. (**C**) In total, 487 TADs were significantly enriched or depleted for CNVs in patients of at least one of the 25 cancer types, including the 79 aforementioned TADs. The heatmap displays the log2 fold-differences to the genomic expectation for those TADs. Rows and columns are clustered based on Euclidean distance using complete linkage.

**Figure 3 cancers-11-01886-f003:**
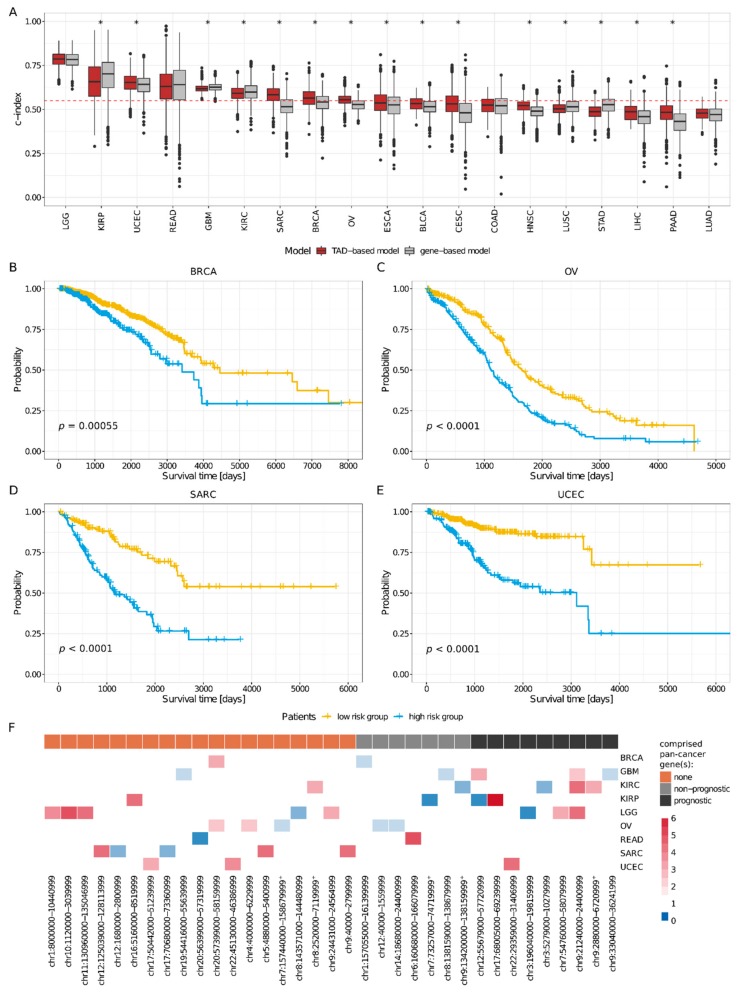
The presence/absence of CNVs in certain TADs is prognostic for overall survival in cancer patients. (**A**) In 9 out of 19 cancer types, the TAD-based models worked, and in four out of those nine, they performed significantly better than the gene-based models, whereas there was no difference in two. The boxplots summarize the c-indices (CI) computed on 1000 random test sets of patients, across 19 cancer types, for the TAD- (red) and the gene-based (gray) models. The dotted red line indicates CI = 0.55. The asterisk (“*”) indicates that the difference between the TAD- and gene-based model is significant at the level of 0.05 (Wilcoxon rank-sum test). A c-index = 1 indicates a perfect prediction, while a c-index = 0.5, a random prediction. (**B**–**E**) Patients were separated into high- (blue) and a low-risk (yellow) groups according to the prognostic features of the final LASSO Cox regression model, subjected to Kaplan–Meier analysis, and compared using the log-rank test, for which the *p*-value (*p* ) is shown. (**B**) BRCA; (**C**) OV; (**D**) SRAC; (**E**) UCEC. (**F**) Thirty-five TADs enriched for CNVs were associated with higher/lower overall survival and, thus, prognostic. The heatmap shows the hazard ratios (HZ) derived from the final LASSO Cox regression model for each prognostic TAD and cancer type; red indicates HZ > 1 (lower survival) and blue HZ < 1 (higher survival). Prognostic TADs were further categorized into three groups: (1) TADs that did not comprise any pan-cancer genes (orange); (2) TADs that comprised only non-prognostic pan-cancer genes (gray); and (3) TADs that comprised prognostic pan-cancer genes (black). Prognostic TADs with a false discovery rate (FDR) greater than 0.05 (see Section 4) are indicated with a plus sign (^+^).

**Figure 4 cancers-11-01886-f004:**
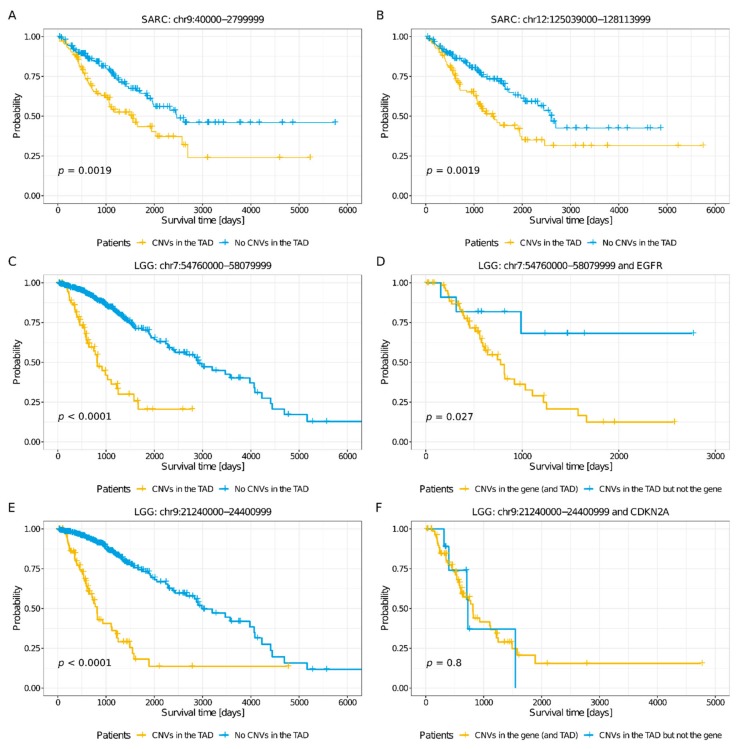
Kaplan–Meier survival analysis for SARC and LGG patients. Time is indicated in days. Patient groups were compared using the log-rank test, for which the *p*-value (*p*) is shown. (**A**,**B**) LASSO Cox regression models identified five prognostic TADs for SARC. Patients were separated into two groups, according to the presence (yellow) or absence (blue) of CNVs in two of these TADs: (**A**) chr9:40000-2799999 and (**B**) chr12:125039000-128113999. (**C**,**D**) LGG patients were separated into two groups, according to the presence or absence of CNVs in the prognostic TAD chr7:54760000-58079999 and in the prognostic pan-cancer gene EGFR, comprised by this TAD. (**C**) Patients with CNVs in chr7:54760000-58079999 (yellow) compared to those without CNVs in this TAD (blue). (**D**) Patients with CNVs in *EGFR* (yellow) compared to those with CNVs in chr7:54760000-58079999 but not affecting *EGFR* (blue). Patients with CNVs in *EGFR* exhibited lower survival compared to those with CNVs in the TAD that did not affect the gene. (**E**,**F**) LGG patients were separated into two groups, according to the presence or absence of CNVs in the prognostic TAD chr9:21240000-24400999 and in the prognostic pan-cancer gene *CDKN2A*, comprised by this TAD. (**E**) Patients with CNVs in chr9:21240000-24400999 (yellow) compared to those without CNVs in this TAD (blue). (**F**) Patients with CNVs in *CDKN2A* (yellow) compared to those with CNVs in chr9:21240000-24400999 in the TAD that did not affect *CDKN2A* (blue). The survival of patients with CNVs in *CDKN2A* did not differ from that of patients with CNVs in chr9:21240000-24400999 that did not affect *CDKN2A*; and both groups exhibited lower survival than the patients without CNVs in this TAD. Hence, CNVs in the prognostic TAD, independently of whether they affect the gene sequence or not, were associated with lower survival of LGG patients.

**Figure 5 cancers-11-01886-f005:**
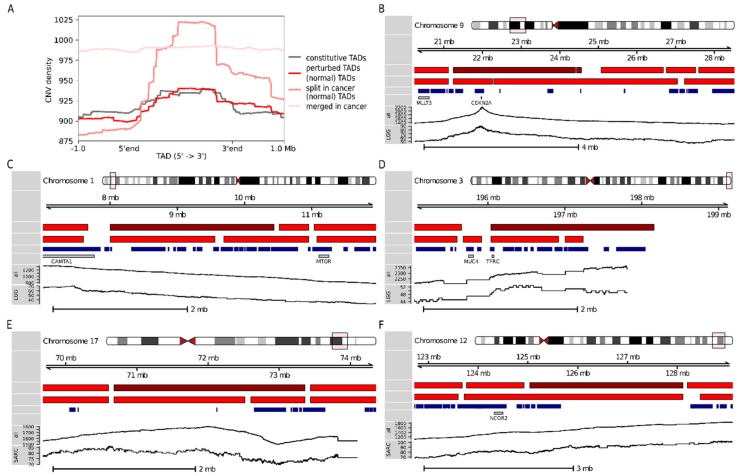
The presence of CNVs tends to be associated with TAD perturbations. (**A**) Density of CNVs calculated for 10-kb-long bins along TADs and 1-Mb-long upstream and downstream genomic regions. TADs were scaled to 2 Mb. The 505 perturbed TADs (dark red) show no difference compared to the 643 constitutive TADs (gray). However, the two subsets of perturbed TADs deviate from the general trend. (**B**–**F**) Genomic context for 35 prognostic TADs. The first two tracks show the physical location on the chromosome and the coordinates of the displayed region. The third and fourth tracks represent the TADs in the normal and cancer genomes, respectively; the TADs in the genomic region under consideration are shown in red, with the TAD(s) of interest highlighted in dark red. The fifth and sixth tracks indicate protein-coding genes and pan-cancer genes, respectively. The sixth track visualizes the number of cancer patients with CNVs at each nucleotide of the genomic region under consideration; the last track restricts patients to those of the cancer type for which the TAD of interest was prognostic. (**B**) Prognostic TADs for LGG chr9:21240000-24400999 and chr9:24431000-24564999 (displayed region: chr9:20240000-28519999); (**C**) prognostic TAD for LGG chr1:8000000-10440999 (displayed region: chr1:7000000-11959999); (**D**) prognostic TAD for LGG chr3:196040000-198159999 (displayed region: chr3:195040000-199159999). (**E**) Prognostic TAD for SARC chr17:70680000-73360999 (displayed region: chr17:69680000-74360999) and (**F**) prognostic TAD for SARC chr12:125039000-128113999 (displayed region: chr12:122719000-129113999).

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
