# Peer review of "Cancer Is Associated with Alterations in the Three-Dimensional Organization of the Genome"

_cancers, 2019, doi:10.3390/cancers11121886_

Round 1
Reviewer 1 Report
The article presents a novel and interesting concept of analyzing CNVs in topologically associating domains as a prognostic biomarker that could be of high significance and represent a novel method for modeling prognosis. However, the manuscript makes several assumptions that limit the interpretation of the data sets and needs some additional control analysis to validate that the results are meaningful.
Major comments:
1) Do the authors have evidence that topologically associating domains are relevant in each cancer tissue as it looks like the majority of TAD data was generated from the corresponding normal tissue?
2) Can they show evidence that TADs are relevant to RNA expression, perhaps through RNAseq data analysis of a correlation of CNV data with expression of genes adjacent to topologically associating domains?
3) The manuscript would benefit from analysis of additional independent data sets, and a deeper dive on clinical variables for a single disease.
4) Can the authors add detail to manuscript to describe the multiple testing that needs to be performed to correct P-values associated with each test? As written, I am concerned that no bonferroni or FDR based corrections were performed in many of the tests invalidating the cox-model interpretation and potentially with the exception of some pathway analysis.
Reviewer 2 Report
This is a review of a manuscript titled "Cancer is associated with alterations in the 3-D organization of the genome" by Li et al.
The paper of Li et al. describes the results of whole-genome study of topologically associating domains (TADs) in various human cancer by using publicly available TCGA data. Author started with building Hi-C based consensus map of TADs based on 24 human tissues and 11 cancer cell lines. By comparing normal TADs with cancer TADs, these TADs were divided into constitutive and perturbed. On the other hand, TADs were classified into enriched or depleted for CNVs, and also each TAD was assessed for presence of pan-cancer genes. Various outputs of this analysis have intrinsic values and may be used as general reference concerning chromosome and locus-level regulation in human genome.The most important analytic outcome of this paper, however, is an identification of a set of prognostic TADs for many different histotypes of human tumors (based on TCGA data). Kaplan-Meier based survival analysis showed that TAD-based models perform at least as good as gene expression based models, and, in many cases, even better, thus, highlighting importance of CNVs and, speaking generally, mutational or epigenetic events affecting non-coding regions. Recent GWAs and sequencing studies many non-coding variants associated with propensity to develop certain cancers or with the course of disease or with its outcomes. Thus, this paper present a resource for prioritizing these non-coding variant, and also highlight the set of "interesting cases" ready for experimental dissection.
On a down-side, this paper present something which certainly is not a "final" dataset of perturbed TADs (the Hi-C set is too small). But for now (2019) - this set is certainly good enough. I recommend accepting this paper.
Reviewer 3 Report
The authors constructed consensus topologically associating domains (TAD) and associated them to CNA in the TCGA series to identify association with prognosis.
Overall the analysis is original and only missing some details on methodologies and discussions.
It seems that the consensus TAD cover 2/3rds of the genome. It seems that after removing the regions that are unnmappable, the entire analyzable genome is basically annotated as TAD, in which case, the authors are simply analysing the TCGA CNA data. The authors should comment and/or discuss on the large portion of the genome that is annotated as a consensus TAD.
Are there any difference between copy number loss and gain?
The authors used a subset of 100 patients for each series for the analysis of CNA, which is quite sound. However, all available patients should be used for the survival analysis (although it seems to be the case, it should be said).
It would be of great interest to all reads to have access to the defined consensus TAD as defined by the authors.
It would be helpful to illustrate the results around CDKN2A.
Figure 5a is missing labels on x and y.
Reviewer 4 Report
The work presented here on the TADs is particularly interesting and searched. The paper is inspiring to read and seems very accurate. It links both a general approach and relevant examples.
It is of course a pity that the Materials and Methods are in the end because a lot of relevant information is in it.
Some parts remain a little vague.
1) it would be necessary to reposition the type of events that is analysed. Indeed, this reviewer is interested in SNPs (in limited numbers), which are the markers of different cancers. This reviewer does not think that COSMIC data (spot SNPs) match.
2) It is a bit difficult to estimate if the number of cases is sufficient to draw broad conclusions and if biases (redundancy for example) are not present.
3) Final question, are these results usable in medical research / hospital? or is it only basic research.
Round 2
Reviewer 1 Report
The authors present an interesting re-analysis of publicly available data sets through projection of TAD data onto TCGA data. The analysis is overall limited without external validation through the generation of novel data sets to test the hypothesis generated by the in situ analysis.
